# Recent Innovations and Nano-Delivery of Actinium-225: A Narrative Review

**DOI:** 10.3390/pharmaceutics15061719

**Published:** 2023-06-13

**Authors:** Sipho Mdanda, Lindokuhle M. Ngema, Amanda Mdlophane, Mike M. Sathekge, Jan Rijn Zeevaart

**Affiliations:** 1Nuclear Medicine Research Infrastructure (NuMeRI), Steve Biko Academic Hospital, Pretoria 0028, South Africa; amanda.mdlophane@sanumeri.co.za (A.M.); mike.sathekge@up.ac.za (M.M.S.); janrijn.zeevaart@necsa.co.za (J.R.Z.); 2Department of Nuclear Medicine, University of Pretoria, Pretoria 0001, South Africa; 3Wits Advanced Drug Delivery Platform (WADDP) Research Unit, Department of Pharmacy and Pharmacology, School of Therapeutic Sciences, Faculty of Health Sciences, University of the Witwatersrand, 7 York Road, Johannesburg 2193, South Africa; lngema1@jh.edu; 4Johns Hopkins Medicine, Department of Radiation Oncology and Molecular Radiation Sciences, Baltimore, MD 21218, USA; 5Radiochemistry, The South African Nuclear Energy Corporation, Pelindaba, Hartbeespoort 0240, South Africa

**Keywords:** actinium-225, recoiling of daughters, targeted radionuclide therapy, radiopharmaceuticals, nanotechnology

## Abstract

The actinium-225 (^225^Ac) radioisotope exhibits highly attractive nuclear properties for application in radionuclide therapy. However, the ^225^Ac radionuclide presents multiple daughter nuclides in its decay chain, which can escape the targeted site, circulate in plasma, and cause toxicity in areas such as kidneys and renal tissues. Several ameliorative strategies have been devised to circumvent this issue, including nano-delivery. Alpha-emitting radionuclides and nanotechnology applications in nuclear medicine have culminated in major advancements that offer promising therapeutic possibilities for treating several cancers. Accordingly, the importance of nanomaterials in retaining the ^225^Ac daughters from recoiling into unintended organs has been established. This review expounds on the advancements of targeted radionuclide therapy (TRT) as an alternative anticancer treatment. It discusses the recent developments in the preclinical and clinical investigations on ^225^Ac as a prospective anticancer agent. Moreover, the rationale for using nanomaterials in improving the therapeutic efficacy of α-particles in targeted alpha therapy (TAT) with an emphasis on ^225^Ac is discussed. Quality control measures in the preparation of ^225^Ac-conjugates are also highlighted.

## 1. Introduction

Advancements in trivalent actinides chemistry have enabled scientists to develop promising radio diagnostic and radiotherapeutic agents for application in nuclear medicine [1]. Yamane et al. (1983) first reported on the radiopolarographic reduction of actinium (III) in aqueous solutions, and demonstrated that the addition of 1, 4, 7, 10, 13, 16-hexaoxacyclooctadecane (18-CROWN-6) to the aqueous solution caused a significant shift in the half-wave potential. The shift was explained by the complex formation of actinium (II) with 18-CROWN-6. The ionic radius and the electronic configuration of Ac^2+^ were determined to be 1.25 Å and [Rn]6d^1^, respectively [2]. Except for actinium (III), the early-to-mid-actinides in the trivalent oxidation state still have a fundamental understanding significantly less developed than that of the main group, the transition metals and lanthanides [1].

Actinium (Ac) occurs naturally from uranium radionuclides (U). The ^225^Ac can be obtained by decaying ^233^U or by neutron transmuting radium-226 (^226^Ra) via successive n, γ capture decay reactions via ^227^Ac, and thorium-228 (^228^Th) to ^229^Th [3]. The radiochemical extraction from ^229^Th has been the main method of producing ^225^Ac, since the early 1990s, with ^229^Th sources (half-life = ~7917 years) obtained by separation from aged, fissile ^233^U [4]. Additional approaches for large-scale production of ^225^Ac have been devised to meet the global demand, and these include the spallation of highly energetic protons onto ^232^Th or U targets, as well as the irradiation of ^226^Ra targets with protons, deuterons, or gamma rays [4]. ^225^Ac produces six predominant radionuclide daughters in the decay cascade to stable bismuth-209 (^209^Bi) [3]. ^225^Ac with a half-life of 10.0 d and particle energy of 6 MeV decays into two high-energy gamma emissions, of which the ^213^Bi 440 keV emission has been used to image drug distribution [3]. ^225^Ac is regarded as an excellent isotope for targeted alpha therapy (TAT) in managing cancer because of its half-life, which allows for sustainable delivery at the tumour site [5]. ^225^Ac and its daughter ^213^Bi are used to label molecules for targeted alpha therapy [6]. Alpha particles typically have a range of 40–90 μm in tissue, sufficient to cover the dimensions of a typical vessel within a tumour and deposit a large amount of energy in a small area, reducing bystander damage [7]. The emitted alpha particle(s) can cause significant radiation damage due to their high linear energy transfer, but their short range causes less damage to the surrounding tissues. On the other hand, a beta particle would have a range of 1–10 mm and would deposit a greater amount of energy into normal tissue [7]. The ^225^Ac radionuclide daughters escape and undergo the process of circulation through the body and accumulation in different organs (especially in the kidneys), which leads to detrimental side effects on healthy organs. Currently, renal toxicity from released ^213^Bi is the major concern [8]. Consequently, there is a consensus to find alternative approaches, such as nano-vehicles to contain the recoiling effect of ^225^Ac daughters, and afford specific deposition of the radionuclides at targeted sites. Interestingly, the application of nanoparticles in the medical field is rapidly expanding, with numerous nano-vectors being developed to facilitate targeted therapy, particularly in cancer management [9]. Moreover, nanoparticles are currently explored in TAT to primarily: (i) reduce the release of radioactive daughters from targeting vectors, (ii) overcome a lack of appropriate ligands for the effective binding of particle emitters to targeting ligands, (iii) reduce distribution to off-target areas, and (iv) alter radionuclide biodistribution [10]. Furthermore, there is a growing interest in investigating ^225^Ac’s chemical binding properties to help with chelator design [11].

Nanotechnology presents endless possibilities in nuclear medicine (i.e., the administration of unsealed radioactive substances to patients in order to perform specific diagnostics and treatments with radiopharmaceuticals) [12]. The combination of nanomaterials and nuclear medicine isotopes opens the door to more precise and effective radiopharmaceuticals. Radiopharmaceuticals have proven to be effective agents due to their ability to be used for diagnostics and therapy [13]. Meanwhile, it remains imperative to consider and address some limitations when developing a radiopharmaceutical, such as its retention by unintended organs, which can affect patient safety, imaging findings, diagnostic accuracy, and therapeutic efficacy [14]. Since the effective use of relevant radionuclides in pre-clinical and clinical studies is dependent on the selection of an appropriate delivery platform [13], this review focuses on recent global efforts to use radioactive decay from ^225^Ac as a promising anticancer therapeutic agent and its improved nano-delivery systems. Moreover, we concisely highlight important clinical studies with ^225^Ac- and ^213^Bi-labelled compounds.

## 2. Targeted Radionuclide Therapy in Cancer Management

Essentially, targeted therapy has seen notable success over the years as an alternative treatment modality for cancer, with the introduction of various therapeutics approved by the US Food and Drug Administration (FDA) that have added new options to the current cancer therapy arsenal of surgery, chemotherapy, and external beam radiation. Monoclonal antibodies and small molecules (i.e., tyrosine kinase inhibitors) are among the known FDA-approved targeted therapeutics for cancer management [9]. Likewise, targeted radionuclide therapy (TRT) is rapidly growing as a class of cancer treatment modalities, with the distinct advantage of eliciting therapeutic action on all cancer sites through selective tumour uptake and retention [15]. TRT employs a targeting vector with selective binding and preferential affinity to tumours to deliver radionuclides which are taken up and retained in tumours or the tumour microenvironment (TME) for effective therapeutic action [16]. Common targeting vectors employed in TRT include antibody (Ab) platforms, peptides, proteins, and small molecules [15].

The selection of radionuclides in TRT mainly includes radionuclides that emit α and β particles and Auger electrons [15,16,17]. A heterogenous radiation dose is delivered through TRT within the tumours at continuously low absorbed dose rates with a radionuclide-dependent variable linear energy transfer (LET) [15]. Presented in Figure 1 is the general depiction of the concept of TRT as employed in cancer treatment [18]. Numerous studies have reported positive therapeutic outcomes from TRT treatments, indicating that TRT prolongs the survival of patients presenting with various cancers, such as metastatic lymphoma, liver, prostate, and thyroid [19,20,21,22]. Meanwhile, Majokwska-Pilip et al. (2020) reported that α-particle-emitting radionuclides used for TAT include ^223^Ra, ^224^Ra, astatine-211 (^211^At), ^225^Ac, lead-212 (^212^Pb), ^227^Th, ^212^Bi, and ^213^Bi. Table 1 summarizes the application of some ^225^Ac and ^213^Bi-labelled compounds in TAT [23].

The application of TRT has significantly progressed over the years, since the memorable approval of Radium ^233^Ra dichloride (Xofigo^®^, Berlin, Germany) by the FDA in 2013, for the treatment of bone metastases from castrate-resistant prostate cancer with no visceral metastases [29]. Accordingly, the prominence of TRT has been further propelled by the recent FDA approvals of ^177^Lu-DOTATATE (Lutathera^®^, Basel, Switzerland) and ^177^Lu-PSMA-617 (Pluvicto™, Basel, Switzerland) for the treatment of neuroendocrine tumours and metastatic castrate-resistant prostate cancer, respectively [30,31]. More studies are underway to advance the scope of TRT application and the understanding of TRT radiobiology. Contrary to external beam radiation therapy (EBRT), the comprehensive understanding of the radiobiology of TRT still falls short [15,19,32]. The heterogenous dose distribution of TRT reportedly renders the absorbed doses by whole organs ineffective in predicting organ toxicity. Moreover, the historically non-standardized usage of patient-specific dosimetry for TRT has made it challenging to ascertain the biological implications of the absorbed dose from TRT [15].

## 3. Recent Investigations on Actinium-225 as a Prospective Therapeutic Agent

### 3.1. Pre-Clinical Studies

The α-emitting radionuclide ^225^Ac possesses nuclear properties that make it very promising for use in targeted TAT, a therapeutic strategy that uses α-particle emissions to destroy tumours. Presented in Figure 2 is the overview of the decay cascade of ^225^Ac with its corresponding radionuclide daughters [33]. Owing to its 10.0 d half-life, rapid decay protocol to sTable 2^09^Bi, and enormous α-particle emitting energies, ^225^Ac is considered a promising candidate for use in cancer therapy [34]. Accordingly, this section provides a concise overview of the prominent ^225^Ac-conjugated agents that have been investigated for application in TAT at a preclinical level. This overview will give insights into the importance of ^225^Ac in TRT and better guide researchers in formulating more efficacious ^225^Ac-based targeted therapeutics to further advance the application of TRT in cancer management.

Targeting the prostate-specific membrane antigen (PSMA) with ^177^Lu-labeled PSMA-specific tracers has emerged as a very viable therapeutic treatment for prostate cancer, according to Ruigrok et al. (2022) [35]. Further, it is reported that the efficacy of this therapy could be improved by replacing the β-emitting ^177^Lu with the α-emitting ^225^Ac. ^225^Ac is believed to have a higher therapeutic efficacy due to the high LET of the emitted α-particles, which can increase the amount and complexity of the therapy-induced DNA double-strand breaks [35]. According to Scheinberg et al. (2011), α-particle-emitting isotopes are being studied in radioimmunotherapeutic applications due to their unparalleled cytotoxicity when targeted to cancerous cells and their relative lack of toxicity towards untargeted normal tissue [34]. Fundamentally, the following are the key properties of the α-particles produced by ^225^Ac: (i) a few cell diameters of tissue range, (ii) high LET resulting in dense radiation damage along each alpha track, (iii) a ten-day half-life, and (iv) four net alpha particles emitted per decay [34]. These allow for a sustained, high radiation dose deposition within the tumour while sparring surrounding healthy tissue [33,34].

Borchardt et al. (2003) investigated intraperitoneal radioimmunotherapy in a mouse model of human ovarian cancer using a targeted ^225^Ac in vivo particle generator with robust selective cytotoxicity while providing a feasible half-life for tumour delivery [36]. It was discovered that the intraperitoneal administration of a ^225^Ac-labeled internalized anti-HER-2/neu antibody significantly extended survival in a nude mouse model of human ovarian cancer at levels that cause no obvious gross toxicity [36]. Moreover, Ballangrud et al. (2004) reported that trastuzumab, a humanized monoclonal antibody directed against HER2/neu, effectively treated breast cancer malignancies [37]. The study tested the efficacy of trastuzumab labelled with the particle emitting atomic generator ^225^Ac against breast cancer spheroids with varying levels of HER2/neu expression. It was concluded that ^225^Ac-labeled trastuzumab could be a potent therapeutic agent against metastatic breast cancer cells with intermediate to high HER2/neu expression [37].

Reissig et al. (2021) elaborated that in nuclear medicine, therapeutic methods such as ionizing radiation are increasingly used to destroy tumours that cannot be treated adequately by other methods such as surgery or chemotherapy [25]. In a mission to find a universal chelator that could allow sensitive bio(macro)molecules to attach and allows ^225^Ac-radiolabeling under mild conditions, the researchers developed an aza-macrocycle-derived mcp chelator with functional groups for universal connection of biomolecules via convenient click chemistry and ^225^Ac-labeling. The ^225^Ac-radioconjugates produced were tested in vitro and in vivo, revealing a high receptor affinity on tumour cells as well as a high tumour accumulation in tumour-bearing mice [25]. In another study, Qin et al. (2020) reported that the overexpression of cholecystokinin B receptor (CCKBR) in human cancers led to the development of radiolabelled minigastrin analogues for targeted radionuclide therapy, which aims to deliver cytotoxic radiation specifically to cancer cells [38]. The cellular uptake and cytotoxic effects of ^225^Ac-labelled and HPLC-purified minigastrin analogue [^225^Ac]Ac-PP-F11N were characterized in the human squamous cancer A431 cells transfected with CCKBR. They further synthesized ^225^Ac-crown-αMSH, with a peptide targeting the melanocortin 1 receptor (MC1R), specifically expressed in primary and metastatic melanoma. It was discovered that the biodistribution of ^225^Ac-crown-αMSH exhibited favourable tumour-to-background ratios at 2 h post-injection in a preclinical model [38].

De Saint-Hubert et al. (2020) further emphasized current achievements and challenges in TAT translational dosimetry [39]. The emphasis was on ^225^Ac and its characteristics, such as limited range within tissue and high LET, which make α-particle emissions more effective in the targeted killing of tumour cells than β-radiation [33,39]. The preclinical biodistribution and dosimetry studies on TAT agents, specifically ^225^Ac and its multiple progeny, and their potential role in better characterizing the pharmacokinetic profile of TAT agents were also highlighted [39]. Lastly, Deblonde et al. (2018) reported that ^225^Ac-based therapies could revolutionize cancer medicine but remain tantalizing due to the difficulties in studying and limited knowledge of ^225^Ac chemistry [40]. They demonstrated a straightforward strategy to purify medically relevant radiometals, actinium(III) and yttrium(III), examined their chemistry using lanmodulin, and concluded that the lanmodulin-based approach charts a new course to study elusive isotopes and develop versatile chelating platforms for medical radiometals, both for separations and potential in vivo applications [40].

Reissig et al. (2022) recently reported on the ^225^Ac-PSMA targeted radioconjugates that have been studied for TAT of metastatic castration-resistant prostate cancer [41]. They further reported that two ligands, mcp-M-alb-PSMA and mcp-D-alb-PSMA, were synthesized by combining a macropa-derived chelator with either one or two lysine-ureido-glutamate-based PSMA- and 4-(p-iodophenyl)butyrate albumin-binding entities using multistep peptide-coupling chemistry and labelled with [^225^Ac]Ac^3+^. The biodistributions of both ^225^Ac-radioconjugates were investigated, and an enhanced binding to serum components in general and to human serum albumin was revealed for [^225^Ac]Ac-mcp-M-alb-PSMA and [^225^Ac]Ac-mcp-D-alb-PSMA [41].

### 3.2. Clinical Studies

Alpha emitters such as ^225^Ac-conjugates can treat cancer more successfully than most radionuclides [33]. Due to the limited range of α-radiation in human tissue (<0.1 mm), which corresponds to average cell diameters, this makes it possible to target cancer cells specifically while protecting the healthy tissues around them [18,34]. In addition, the strong linear energy transfer brought on by the high energy of α-particles results in a notable increase in the number of cells killed. Because of this, α-radiation is a viable treatment option for tumours resistant to traditional therapies [35]. Furthermore, the alpha emitters ^225^Ac and ^213^Bi are promising therapeutic radionuclides for use in TAT for cancer and infectious diseases [33], and according to Morgenstern et al. [42], there has been an ongoing interest in ^225^Ac and bismuth due to their interesting chemical and physical characteristics. Various clinical trials have proven the viability, safety, and therapeutic efficacy of targeted alpha therapy using ^225^Ac and ^213^Bi [42]. Shown in Figure 3 is a patient-response depiction from one of the select clinical studies of ^225^Ac [43].

According to Satapathy et al. (2020), ^225^Ac labelled PSMA-617 is a robust treatment option for managing metastatic castration-resistant prostate cancer (mCRPC). A study was carried out to evaluate the impact of ^225^Ac-PSMA-617 therapy on the quality of life of patients with heavily pretreated mCRPC using the NCCN-FACT-FPSI-17 questionnaire from the National Comprehensive Cancer Network. Despite extensive pretreatment and the advanced state of the disease, ^225^Ac-PSMA-617 significantly improved the health-related quality of life for mCRPC patients [44]. This correlated with the data from a comprehensive study by Sathekge M et al. (2019), which revealed that ^225^Ac-PSMA-617 confers remarkable therapeutic efficacy in heavily pretreated mCRPC patients. Seventeen patients were treated, and it was observed that the extraordinary therapeutic efficacy reported could be achieved with less toxicity to salivary glands due to the de-escalation of administered activities in subsequent treatment cycles [43]. Lawal et al. (2022) further reported that ^225^Ac-labeled prostate-specific membrane antigen is safe and effective in treating mCRPC. They highlighted that no study had specifically assessed its safety in patients with extensive skeletal metastases of mCRPC [45]. They then investigated the hematologic toxicity and efficacy of ^225^Ac-PSMA-617 therapy in patients with extensive skeletal metastases of mCRPC. The median progression-free survival and overall survival of the study population were found to be 14.0 months (95%CI: 8.15–19.86) and 15.0 months (95%CI: 12.8–17.2), respectively. ^225^Ac-PSMA-617 induced a marked antitumour effect in ~80% of patients with extensive skeletal metastases of mCRPC with a rare incidence of severe hematologic toxicity. Age, number of treatment cycles, and the presence of renal dysfunction were significant risk factors for hematologic toxicity of ^225^Ac-PSMA-617 therapy [45].

Jurcic et al. (2018) reported that TAT kills tumours more effectively while protecting nearby healthy cells [46]. It was also emphasized that several clinical trials were carried out to evaluate the safety, efficacy, and anti-leukaemic effects of lintuzumab utilizing α-emitters ^213^Bi and ^225^Ac [46]. The phase I study conducted in 18 patients with relapsed or refractory acute myeloid leukaemia demonstrated the safety and antitumour effects of ^213^Bi-lintuzumab and ^225^Ac-lintuzumab. Positive clinical outcomes were reported in phase I trials of ^225^Ac-lintuzumab, and a phase II study of ^225^Ac-lintuzumab monotherapy for older patients with untreated acute myeloid leukaemia is now in progress and is also being studied in a subset of patients with CD33-positive multiple myeloma [46].

Jurcic et al. (2015) had previously reported that ^225^Ac-lintuzumab consisted of a radiometal that emits α-particles linked to an anti-CD33 antibody, and further stated that in patients that received ^225^Ac-lintuzumab and low-dose of cytarabine over weeks period, twelve were treated. It was concluded that a fractionated dose of ^225^Ac-linutuzmab can be combined safely with low-dose cytarabine to confer anti-leukaemic activity [47]. Meanwhile, Finn et al. (2017) reported on a multicenter phase II study evaluating the response rate, progression-free survival and overall survival after fractionated-dose of ^225^Ac-lintuzumab monotherapy in older acute myeloid leukaemia patients unfit for induction chemotherapy [48]. In the study, two doses of 2 μCi/kg of ^225^Ac-lintuzumab were administered on days 1 and 8, granulocyte colony-stimulating factor was administered starting 10 days after the 2nd dose of ^225^Ac-lintuzumab, and spironolactone was given for one year to prevent possible radiation-induced nephrotoxicity. A total of thirteen patients were treated, and a 56% response rate was recorded in older patients unfit for intensive therapy [48]. Rosenblat et al. (2022) further reported that the anti-CD33 antibody lintuzumab has modest activity against acute myeloid leukaemia. They demonstrated that therapy for acute myeloid leukaemia with the targeted α-particle generator ^225^Ac-lintuzumab was feasible with an acceptable safety profile [49].

## 4. Nanodelivery of Radionuclides

Several drugs can be administered effectively and safely using nanoscale-diameter particles such as liposomes, dendrimers, carbon nanotubes and inorganic nanomaterials [9,50]. The benefits of nanosystem-mediated drug delivery include adequate absorption, delayed metabolism and excretion, longer half-life, ability to cross multiple anatomical barriers, and enhanced accumulation in specific target tissues [50]. Nanomaterials are complex multifunctional structures that have the potential to improve the efficacy of traditional cancer treatment methods. The development of novel nanosystems incorporating nuclear entities in delivery nano-vectors (Figure 4) has made significant progress, with the potential to reshape the nuclear medicine direction [51]. Nonetheless, translation from preclinical evaluations to clinical use is lacking, and significant investment is required to achieve this goal [51].

Liposomes have previously been considered for radionuclide diagnostic and therapeutic delivery [54]. Liposomes are drug delivery systems that consist of lipid bilayers enclosing a hydrophilic interior where the active pharmaceutical ingredient accumulates. Other variations of this technology are used with different interactions between hydrophobic and hydrophilic layers, such as oil-emulsion type systems [51]. Due to the high kinetic energy, α-particle will escape the liposomal phospholipid membrane to irradiate the targeted cells [23,51]. Additionally, dendrimer-based nanoparticles have been developed and have shown promising results in many areas [55]. However, there are key factors that must be considered before the construction of radiolabelled dendrimers. For example, the selection of appropriate radionuclides and dendrimers are very important, given the critical role of radionuclide’s half-life and also the physicochemical properties of dendrimer of interest, either for the imaging and treatment of cancer, cardiovascular and other diseases [51,55]. Essentially, to achieve the desired results, the half-lives of selected radionuclides must be compatible with the pharmacokinetic profiles of dendrimers [55]. Dendrimers conjugated with diethylenetriamine pentaacetate (DTPA), for instance, can be easily labelled with ^99m^Tc. DTPA could be used as a ^99m^Tc chelator with a high radiochemical yield and stability [55]. Due to the simplicity of ^99m^Tc radiolabelling in dendrimers, various dual-model imaging applications, such as SPECT/CT, SPECT/MR, and SPECT/optical imaging, have been developed [55]. Criscione et al. (2011) conjugated triiodinated moieties and ^99m^Tc on the surface of G4 PAMAM dendrimers for SPECT/CT application in a previous study [56].

Nonetheless, Woodward et al. (2011) reported that radioisotope leaching remained a challenge in the nanodelivery of radionuclides, resulting in the escape of radionuclide daughters. Consequently, Woodward et al. (2011) developed 3–5 nm diameter monazite (LaPO4) inorganic nanoparticles as carriers for ^225^Ac and approximately or less than 50% of the ^221^Fr and ^213^Bi were retained from the nanoparticle lattice. Furthermore, Rojas et al. (2015) added two layers of LaPO_4_ and reduced leaching of ^221^Fr to 20% [57].

### Nanodelivery of Actinium-225

The main difficulty with α-generator radiotherapies is that traditional chelating moieties cannot sequester the radioactive daughters in the bioconjugate [51]. Various nanostructures that can encapsulate ^225^Ac and retain its daughter constituents at tumour sites have been investigated to mitigate the challenge of undesired cytotoxicity to healthy tissues [58]. Research has demonstrated that a liposomal nanoparticle-antibody conjugate can deliver radionuclides and contain the decay daughters of ^225^Ac while targeting biologically relevant receptors [59]. It was further revealed that the radium and ^225^Ac radionuclides could be loaded into sterically stabilized liposomes in high yields, then after coating the liposomes with a folate-F(ab′)(2) construct, a product with an affinity for tumour cells expressing folate receptors [59]. Researchers have reported that the Multivesicular liposomes (MUVELs) made of different phospholipids may improve ^225^Ac daughter retention [60]. PEGylated MUVELs retained 98% of encapsulated ^225^Ac over time, and it was concluded that MUVELs might be able to deliver higher fractions ^225^Ac into targeted sites [60]. Moreover, stable PEGylated phosphatidylcholine-cholesterol liposomes of various sizes and charges were formulated. The study supported the hypothesis that it may be possible to develop ^225^Ac-based therapies by delivering multiple ^225^Ac particles in liposomes [60,61].

Polymersomes are also emerging as delivery platforms. As such, these nano-vesicles have also been investigated regarding the delivery of ^225^Ac and the recoiling daughters in cancer therapy [58]. de Kruijff et al. (2019) employed polymersomes to encapsulate ^225^Ac, which was injected intratumourally into tumour-bearing mice. The prepared ^225^Ac-encapsulated polymersomes exhibited enhanced antitumour activity with limited renal toxicity [62]. A separate study by de Kruijff et al. (2017) initially reported on the formulation of a ^225^Ac-labelled polymersome nanosystem and evaluated the retention of ^225^Ac daughters (^221^Fr and ^213^Bi) in vitro [63]. It was found that the nanosystem resulted in increased retention of ^221^Fr and ^213^Bi, owing to the high encapsulation efficiency of ^225^Ac into the nano-vesicles. In addition, the nanosystem also demonstrated reduced toxicity capacity [63].

Cdrowska et al. (2018) proposed utilizing titanium dioxide (TiO_2_) nanoparticles as a carrier for ^225^Ac and its decay products [57]. Accordingly, TiO_2_ nanoparticles functionalized with substance P, a peptide fragment which targets NK1 receptors on the glioma cells, were formulated, and it was reported that only after 10 days was a 30% leaching of ^221^Fr recorded. The researchers further stated that the synthesized ^225^Ac-TiO_2_-PEG-SP showed a high cytotoxic effect in vitro in T98G glioma cells and, therefore, could be a promising new radioconjugate for targeted radionuclide therapy of brain tumours [57]. Salvanou et al. (2020) recently reported on a study to evaluate gold nanoparticles radiolabeled with ^225^Ac as an injectable radiopharmaceutical form of brachytherapy for local cancer radiation treatment [52]. Interestingly, the study revealed that ^225^Ac-Au@TADOTAGA resulted in tumour growth inhibition [52]. Meanwhile, Matson et al. (2012) stated that single-walled carbon nanotubes, previously demonstrated as trivalent lanthanide ion and small molecule sequestering agents, successfully incorporate [^225^Ac]Ac^3+^. They added that bath sonication was used to load [^225^Ac]Ac^3+^ ions and gadolinium (Gd^3+^) ions into ^225^Ac@gadonanotubes (^225^Ac@GNTs). The gadonanotubes, which are carbon nanotubes containing superparamagnetic clusters of Gd^3+^ ions, successfully sequestered [^225^Ac]Ac^3+^ ions in the presence of Gd^3+^ ions and retained them after a human serum challenge, making ^225^Ac@GNTs potential candidates for radioimmunotherapy for the delivery of [^225^Ac]Ac^3+^ ions at higher concentrations than current traditional ligand carriers [64].

In 2014, McLaughlin et al. [65] prepared lanthanide phosphate (LnPO_4_)-based nanoparticles coated with GdPO_4_ for targeted α therapy of ^225^Ac in lung cancer. The nanoparticles comprising [^225^Ac]Ac^3+^ were further coated with an additional layer of gold and functionalized with a thrombomodulin-targeting monoclonal antibody (mAb 201b) for targeting lung endothelium. Almost 70% of ^225^Ac daughter radionuclide (^213^Bi) was retained in lung tissues at 1 h post-injection. Additionally, it was observed that the treatment with targeted ^225^Ac nanoparticles demonstrated significantly enhanced anticancer activity compared to the cold mAb. The study concluded that the formulated nanoparticles could deliver ^225^Ac without limiting the therapeutic properties of the α-radiation to tumours [65]. Gadolinium vanadate (GdVO4) core and core +2 shell nanocrystals (NCs) have also been explored in TAT, as well as in magnetic resonance through the formulation of gadolinium doped nanocrystals embedded with ^225^Ac and ^227^Th as α-emitter radionuclides. The nanocrystals were investigated for in vitro retention of parent ^225^Ac and ^227^Th radioisotopes and their corresponding decay daughters and later proposed as promising theragnostic agents for biomedical treatments [66].

Based on the results from the numerous studies highlighted above, the targeted nanodelivery of ^225^Ac presents an elegant approach for achieving maximal uptake in tumours, while limiting the unwanted retention in the liver and spleen, as well as healthy tissues. Moreover, the approach allows for the effective, selective delivery of ^225^Ac to tumours without hampering the therapeutic properties of the radionuclide. Accordingly, in the future, it is envisaged that the use of target-specific polymersomes and liposomal nano-vectors for targeted delivery of ^225^Ac using targeting ligands (i.e., small molecules such as folate, as well as monoclonal antibodies and peptides) that specifically bind to relevant tumour receptors, will advance as a superior alternative modality in TAT. However, additional protocols still need to be explored to optimize the activity of antibodies and peptides, to circumvent some of the limitations, such as limited diffusion capacity and penetration, compared to smaller molecules [59].

## 5. Quality Control and Preparation of Actinium-255 Conjugates

Radiopharmaceuticals are considered regular medicinal products, and as such, they must be manufactured in accordance with the principles and guidelines of good manufacturing practices [67]. The European Pharmacopoeia has created special radiopharmaceutical monographs to address quality control concerns [68]. Thakral. et al. (2021) documented an optimized protocol for preparing therapeutic doses of ^225^Ac-PSMA-617 with high yield and radiochemical purity (RCP). The radiopeptide was obtained with an adequate yield of 85–87% and RCP of 97–99%. The protocol enables single-step, successful, routine in-house radiolabelling of ^225^Ac with PSMA-617 with high yield and RCP [5]. Apostolidis et al. (2005) also reported on a method for separating and purifying ^225^Ac from a ^229^Th source [69]. The method is based on a combination of ion exchange and extraction chromatographic methods in nitric acid media and allows the preparation of carrier-free, clinical grade ^225^Ac with an overall yield exceeding 95%. Radiometric (spectrometry) and mass spectrometric methods are used for product quality control, and the ^225^Ac product can be loaded onto a radionuclide generator to produce ^213^Bi for preclinical and clinical studies of cancer targeted therapy [69].

Abou et al. (2022) investigated the impact of ^225^Ac/^227^Ac material in the radiolabelling and radiopharmaceutical quality control evaluation of a DOTA chelate-conjugated peptide [70]. The radiolabelled products were characterized using thin-layer chromatography, high-pressure liquid chromatography, gamma counting, and high-energy resolution gamma spectroscopy. Peptide was radiolabelled and evaluated at >95% RCP with high yields for generator produced ^225^Ac [70]. Kelly et al. (2021) also showed that the determination of RCP of ^225^Ac-labeled radiopharmaceuticals is an essential quality control measure and later reported on the use of radio thin layer chromatography (radio-TLC) to determine RCP [71]. Likewise, several studies have reported on the use of radio-TLC to identify various radiochemical forms of ^225^Ac-conjugates, including ^225^Ac-PSMA-I&T by Hooijman et al. (2021) for the translation into a clinical phase 1 dose escalation study [72]. Dumond et al. (2022) also stated that the radiochemical purity of ^225^Ac-PSMA-617 was evaluated utilizing radio-TLC. The method provided ^225^Ac-PSMA-617 in high radiochemical yield of >99% and radiochemical purity of 98 ± 1%, formulated for preclinical studies [73].

## 6. Conclusions

The distinct ability of targeted radionuclide therapy (TRT) to achieve selective uptake and retention of radionuclides by tumours is a remarkable attribute which makes this modality a suitable alternative anticancer treatment to conventional therapies. Likewise, the positive outcomes from pre-clinical and clinical investigations on ^225^Ac indicate that this radioisotope holds great promise as a prospective anticancer agent. The recent advancements in nanotechnology provide a great platform to explore and enhance the therapeutic activity of ^225^Ac for application in cancer as a nanomedicine. The promising outcomes from various nano-delivery systems such as liposomes, dendrimers, carbon materials, and inorganic nanomaterials that demonstrate optimal in vitro and in vivo biodistribution and retain the recoiling ^225^Ac daughters could be leveraged on to devise more innovative nano-based ^225^Ac therapeutics. Quality control measures remain critical in preparing radiopharmaceuticals, including ^225^Ac conjugates, and must always be in place to adhere to good manufacturing practices.

## Figures and Tables

**Figure 1 pharmaceutics-15-01719-f001:**
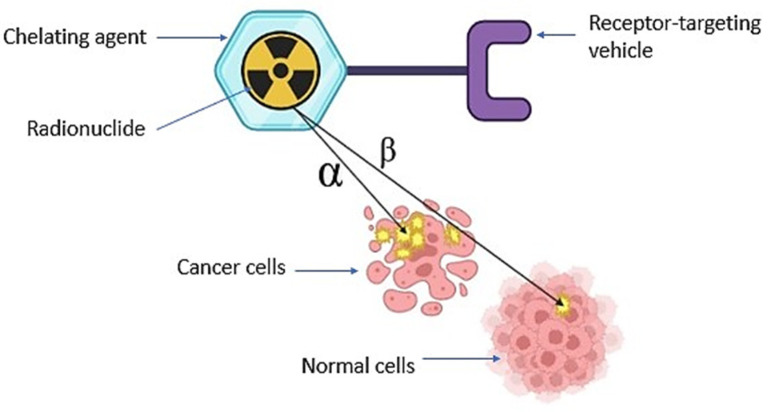
Schematic diagram illustrating the action of either alpha (α) or beta (β) particle emission in targeted therapy.

**Figure 2 pharmaceutics-15-01719-f002:**
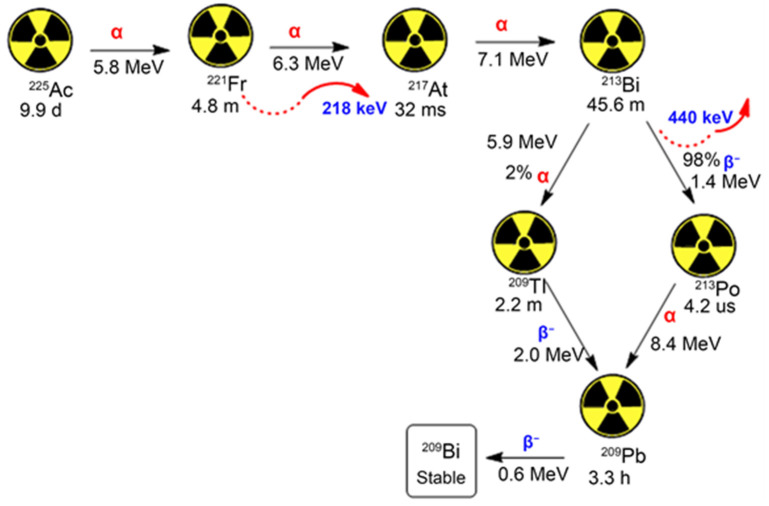
The illustration of ^225^Ac decay via a cascade of six radionuclides to stable 2^09^Bi [33].

**Figure 3 pharmaceutics-15-01719-f003:**
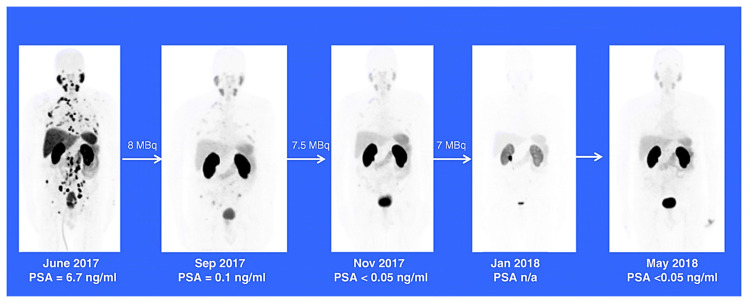
Illustration of ^225^Ac-PSMA-617 treatment in chemotherapy-naive patients with advanced prostate cancer. A patient presenting with advanced prostate cancer exhibited a complete response post 3 treatment cycles with ^225^Ac-PSMA-617. The patient remained symptom-free at an 11-month follow-up, with his serum PSA remaining below the detectable level and the follow-up ^68^Ga-PSMA-11 PET/CT scan remaining negative for disease recurrence [43].

**Figure 4 pharmaceutics-15-01719-f004:**
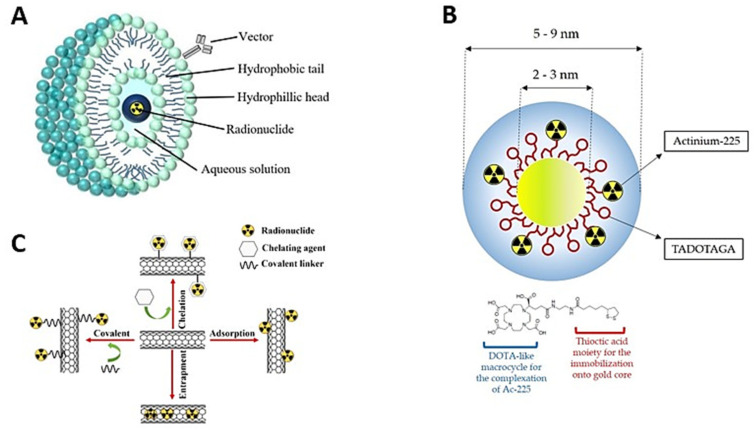
Illustration of various forms of nanostructures for radionuclide nanodelivery. A nanodelivery of choice may be (**A**) liposomal [23], (**B**) gold (Au) decorated nanostructures [52], or (**C**) carbon-based nanotubes [53].

**Table 1 pharmaceutics-15-01719-t001:** Overview of ^225^Ac and ^213^Bi conjugates targeted alpha therapy application.

Cancer Type	Radionuclide	Conjugate	References
LS174T tumours	^225^Ac	HEHA MAb CC49	[24]
Prostate cancer	^225^Ac	PSMA	[25]
Breast, ovarian, lung, gastric and prostate	^225^Ac	HER3	[26]
Bladder cancer	^213^Bi	EGFR	[27]
Ovarian cancer	^213^Bi	sdAbs	[28]

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
