# Peer review of "Recent Innovations and Nano-Delivery of Actinium-225: A Narrative Review"

_pharmaceutics, 2023, doi:10.3390/pharmaceutics15061719_

Round 1

Reviewer 1 Report

The review gives a nice overview of the use of Actinium-225. The story that is told is comprehensive, and built up logically from a general description of TRNT, over TAT with Actinium to the nanoparticle applications within TAT. The review gives a nice overview of the field and includes most relevant papers. At some stages there are some inconsistencies throughout the document which should be remedied. Overall I think this paper is of interest and could be published after implementation of some remarks:

Line 17: rephrase, the subsentence seems to be inserted in the wrong place.

Line 49 to 51: rephrase and remove particle after 255Ac, […] and an alpha particle […]

Line 49 and 52: the half-lives are not the same. One states 9.9 days, the other 10.0

Line 52: replace accumulation by delivery

Line 54-55: resulting in a higher dose compared to what?

Line 57: There is no such thing as a “targeted” alpha particle, or at least it does not have any influence on the range. Remove “targeted”. Additionally, alpha particles do not have a diameter of 50-80 micron. They do have that range though. Please adjust. Additionally, the range is medium dependent, so please specifiy  in which medium the alpha particles have this range.

Line 60: Same issue as comment above. remove “targeted”

Line 61: add “on the one hand” before or remove “on the other hand”

Line 62: Earlier, the authors always use “Ac’s”. now the manuscript shifts to Actinium’s. It would be nice to see consistency throughout the document.

Line 107: Gamma rays are not used for therapy so should not be listed here. It would however be worthwhile to elaborate a bit more on the theranostic side of the radipharmaceutical applications.

Line 115: The TAT acronym has been defined earlier. No need to repeat.

Line 112: Figure 1 represents the general mode of action of the therapeutically relevant radioisotopes, not only Ac-225. Figure caption is not ok.

Line 130-131: The way it is phrased it seems RaCl2 is used for the treatment of prostate cancer. [233Ra]RaCl2 is however used to treat bone metastasis following prostate cancer but not the primary tumor itself. Please amend.

Line 146: See earlier comments on explaining TAT. Only explain an acronym in the first use of the document

Line 148: see earlier comment on half-life. Please use consistent half-life throughout the document.

Line 150: remove space before “Accordingly”

Line 165-166: be consistent with the isotope notation (Lutetium-177 versus Lu-177 versus 177Lu)

Line 173: please explain why these 4 factors are usefull. Specifically the 10 days half-life. Why is this advantageous?

Line 178: replace internalizing by internalized

Line 198: choose between American or British spelling and stick to it for the entire paper

Line 335: change “alpha particle emissions” to “alpha particles”

Line 336: It could be nice to add a representation of dendrimers to figure 4

Line 337: What are the applications of dendrimers you talk about here?

Line 340: I assume the authors mean that the physical half-life should match the biological half life? It would be nice to elaborate on this then.

Line 350: the section on LaPO4 comes a bit sudden and is directly linked to a sentence on dendrimers. This section seems to be a bit misplaced? Or at least I do not see a link between the section and the previous one. Please amend.

Line 362: multiple radiations? Please specify

Line 372: 225Ac radionuclides rather than Ac-Particles

Line 376 and 379: line 376 states the author is “Kruijff” while line 379 states “de Kruijff”. I assume this is the same author so please correct the name. The comment is also for references 61 and 62.

Line 385: decay products and not decomposition products

Line 385: what does substance P targets?

Line 396: correct way of writing this is [225Ac]Ac3+

Line 397: specify what are the gadonanotubes

Line 398: see comment line 396. Check across the entire document.

Line 399: do you really think that CNTs are potential candidates for TRT? They mostly accumulate in the lungs and stay there. Please comment on this

Line 405: There is a typo in what I suppose should be endothelium

Line 412: what type of nanocrystal? What is the composition?

Line 442-444: be consistent with the notation

Remark on the bibliography: check and correct the isotope and molecules notation (ref 4, 5, 7, 18, 27, 30, 32, 37, 42, 44, 46, 47, 56, 61, 62, 63, 64, 65, 67, 68, 70, 71, 72)

·        Overall quality of the English is good. There are some sentences that are sometimes built wrong so they become confusing. Please check the document for consistency follow the concensus nomenclature rules for radiopharmaceutical chemistry (10.1016/j.nucmedbio.2017.09.004)

Author Response

Dear Reviewer.

Thank you for taking your time and review this manuscript. Please find the attached responses. Please note the line numbers manuscript has shifted due to editing of the manuscript draft.

Thank You.

Reviewer 2 Report

- structure of the review is incomplete or illogical, f.i. I miss a paragraph of the production of Ac-225 and/or its physical features of this radioiosotope (now, it is mentioned in some other paragraphs), further, in the section itself, 'preclinical and clinical studies', I miss a structure or a clearly overview, what has been done in the past (f.i. a table or so)

- choice of words is not good, f.i. ...TRT as an alternative anticancer intervention..., better; TRT as an alternative for cancer treatment,

other; 132; castrattio-...is wrong; castrate-..., line 153 aide, better guide...

- choice of words is not good, f.i. ...TRT as an alternative anticancer intervention..., better; TRT as an alternative for cancer treatment,

other; 132; castrattio-...is wrong; castrate-..., line 153 aide, better guide...

Author Response

Dear Reviewer.

Thank you for taking your time and review this manuscript.

Please find the attached response.

Thank You.

Reviewer 3 Report

The authors present an up-to-date review of targeted alpha therapy with Ac-225 and its daughter Bi-213. I am not aware of a similar recent review. Following a concise review of preclinical and clinical studies, the authors then address progress in improvements of binding chemistry for delivery and retention of the payload as its chemistry changes through the decay chain. This is probably the most important contribution of the manuscript. Finally, the authors briefly address challenges in preparation and quality control of these radioconjugates.

MINOR
The manuscript could do with a bit of rewriting for consistent flow, rather than returning to topics and repeating information. For example, the first para on page 5 repeats earlier discussion of advantages of alpha over beta therapeutics. These advantages are repeated again on page 6 in the first para of Clinical Studies.

Page 6, Figure 3 caption. Must give more information. I presume these are Ga-68 PSMA scans.

TYPOS ETC

Page 1, line 17. I don’t think 225Ac should have a possessive

Page 2, line 49. Is the use of the term “particle” correct here?

Page 2, line 52. Here it states 9.9 day half life where line 49 says 10.0 days

Page 2, lines 52-55. Doesn’t this last sentence restate what was said at the top of the page?

Page 2, line 56. You defined “targeted alpha therapy” as “TAT” in line 51. Why not use TAT here?

Page 2, line 62. Possessives again

Page 2, line 79. Should remove “and radiotracers” because all radiotracers used in nuclear medicine are radiopharmaceuticals

Page 2, line 85. Suggest changing “diagnostic and therapeutic accuracy” to “diagnostic accuracy and therapeutic efficacy”

Page 4, line 134. Typo – “castration”

Page 4, line 148. Similar discrepancy between 10.0 day half life in text and 9.9 day half life in Figure 2

Page 13, ref 36. Typo - first author’s initials AM, not AsM

There are inconsistencies in the formatting of the reference list

Author Response

Dear Reviewer,  

Thank you for taking your time and review this manuscript.

Please find the attached responses.

Thank You
